# Effect of Micronutrient Powder (MNP) with a Low-Dose of Iron on Hemoglobin and Iron Biomarkers, and Its Effect on Morbidities in Rural Bangladeshi Children Drinking Groundwater with a High-Level of Iron: A Randomized Controlled Trial

**DOI:** 10.3390/nu11112756

**Published:** 2019-11-13

**Authors:** Sabuktagin Rahman, Patricia Lee, Rubhana Raqib, Anjan K. Roy, Moududur R. Khan, Faruk Ahmed

**Affiliations:** 1Public Health, School of Medicine, Griffith University, Gold Coast Campus, QLD 4220, Australia; 2International Centre for Diarrhoeal Disease Research, Bangladesh, Mohakhali Dhaka 1212, Bangladesh; 3Institute of Nutrition and Food Science, University of Dhaka, Dhaka 1000, Bangladesh; khan.moudud@gmail.com; 4Menzies Health Institute Queensland, Griffith University, Gold Coast, Queensland 4222, Australia

**Keywords:** Micronutrient Powder, groundwater iron, hemoglobin, morbidity, children, Bangladesh

## Abstract

Micronutrient Powder (MNP) is beneficial to control anemia, but some iron-related side-effects are common. A high level of iron in the groundwater used for drinking may exacerbate the side-effects among MNP users. We conducted a randomized controlled trial examining the effect of a low-dose iron MNP compared with the standard MNP in children aged 2–5 years residing in a high-groundwater-iron area in rural Bangladesh. We randomized 327 children, who were drinking from the “high-iron” wells (≥2 mg/L), to receive either standard (12.5 mg iron) or low-dose iron (5.0 mg iron) MNP, one sachet per day for two months. Iron parameters were measured both at baseline and end-point. The children were monitored weekly for morbidities. A generalized linear model was used to determine the treatment effect of the low-dose iron MNP. Poisson regressions were used to determine the incidence rate ratios of the morbidities. The trial was registered at ISRCTN60058115. Changes in the prevalence of anemia (defined as a hemoglobin level < 11.0 g/dL) were 5.4% (baseline) to 1.0% (end-point) in the standard MNP; and 5.8% (baseline) to 2.5% (end-point) in the low-dose iron MNP groups. The low-dose iron MNP was non-inferior to the standard MNP on hemoglobin outcome (β = −0.14, 95% CI: −0.30, 0.013; *p* = 0.07). It resulted in a lower incidence of diarrhea (IRR = 0.29, *p* = 0.01, 95% CI: 0.11–0.77), nausea (IRR = 0.24, *p* = 0.002, 95% CI: 0.09–0.59) and fever (IRR = 0.26, *p* < 0.001, 95% CI: 0.15–0.43) compared to the standard MNP. Low-dose iron MNP was non-inferior to the standard MNP in preventing anemia yet demonstrated an added advantage of lowering the key side-effects.

## 1. Introduction

Anemia is a major public health problem in the low- and middle-income countries [1]. Anemia in children, defined as a hemoglobin level < 11.0 g/dL, is associated with impaired cognitive performance; increased mortality and morbidity; and poorer educational attainment in children [2]. Iron deficiency (ID) is considered as the most common cause of anemia, with the widely held assumption that half of all anemia cases are caused by ID [3]. The World Health Organization recommends Micronutrient Powder (MNP), a powdered formulation consisting of key micronutrients, including iron, as an intervention to prevent childhood anemia [4]. Accordingly, the Bangladesh Government has also adopted this intervention to prevent childhood anemia. However, an increasing number of studies have shown that the supplementation of iron/MNP is associated with side effects, such as diarrhea, nausea, vomiting, bloody stool, malaria, and respiratory tract infections [5,6,7]. Of note, iron is a pro-oxidant and can have deleterious effects if an excessive amount of free iron is present in the body system [8]. Iron in the body is maintained by a tightly controlled regulatory system, and uptake of iron in the body depends on the iron status of the body, and/or the presence of inflammation and infection. In the presence of a sufficient reserve of body iron or systemic inflammation, the intestinal uptake of iron may be limited due to hepcidin-mediated regulation [9,10]. The unabsorbed iron in the gut might affect the composition of the gut microbiome, leading to the side effects [11,12]. In this context, trials have been conducted assessing the efficacy of low-iron MNPs in African settings with a high infection burden. Findings have shown that despite there being efficacy with the low-iron formulations in improving hemoglobin levels, increased side-effects were documented compared to placebo [13,14]. Groundwater iron has been an evolving area of research in anemia science [15,16]. Iron is one of the most abundant metals on Earth, and is ubiquitous in groundwater sources depending on the environment over which the water flows [17,18]. Recent studies have shown a significant association between iron status and daily iron intake from drinking groundwater in different population groups [15,19]. Further, iron status was observed to be good in Bangladeshi populations who are drinking from groundwater with a high level of iron [15,18,19,20]. In the country, the MNP program for the prevention of childhood anemia suffers poor coverage (~2%–3%, personal communication), and the side-effects are documented [21]. To date, no study has been conducted to examine the usage of MNPs/iron supplements in iron-replete children, whose potable supplies are iron-rich groundwater. Hence, the present study examined the effect of a low-iron MNP compared with the standard MNP on the hemoglobin concentration, and the associated morbidities, in Bangladeshi children exposed to high-iron groundwater.

## 2. Materials and Methods 

### 2.1. Study Design, Participants and Randomization

A randomized controlled trial was conducted among children, 2 to 5 years of age, in the Belkuchi sub-district in north-western Bangladesh. Belkuchi is located within the high-iron groundwater areas [17] and all enrolled children were reported to drink groundwater with a high level of iron. Of note, an iron concentration ≥ 2 mg/L was considered as high based on the cut-off for the tolerable upper limit of iron in water defined by the Joint FAO/WHO Expert Committee on Food Additives (JECFA) [18,22]. The exclusion criteria were children receiving MNPs/iron supplements and/or antibiotics in the preceding two months; the presence of chronic, congenital debilitating illnesses; and the guardian’s unwillingness to participate. A total of 327 children were randomly allocated to receive either a low-dose-iron MNP (containing 5 mg Fe, 300 µg RE vitamin A, 5 mg zinc, 30 mg vitamin C, and 0.15 mg folic acid) or a standard MNP (containing 12.5 mg Fe, 300 µg RE vitamin A, 5 mg zinc, 30 mg vitamin C, and 0.15 mg folic acid) to consume 1 sachet every day for 60 days. It is important to note that the standard MNP formulation has been recommended by the Bangladesh Government for the prevention and control of anemia in children and, accordingly, there is a significant distribution of MNPs by national NGOs. Thus, we did not consider a placebo arm due to ethical concerns. Randomization was done at two levels. At first, roughly 70% of the total enrolled children (*n* = 327) were selected by simple random sampling using a random number generator for collecting blood samples. In the second step, randomization was carried out by an independent researcher and the children were allocated to one of the two letter codes (A and B) using a random number generator, without allowing for duplicate entries and not fixing a seed [23]. The sachets containing MNP preparations (Standard MNP and low-iron MNP) were identical in appearance. The sachets were labelled by the manufacturer (Manisha Pharmoplast Pvt. Ltd, Gujarat, India) with alphabetic codes (A and B) for group identification. The MNP preparations were analyzed for a quality control check by the manufacturer and the amounts of all ingredients were found within required ranges. Except for one (SR), all the investigators, field personnel and participants were blinded to the group assignment. The codes were not disclosed to the researchers until preliminary analysis was completed. The purpose and exact nature of the study were explained to the mothers or caregivers of all prospective participants. We further explained that the project physician would help to manage if their children encounter any common side-effects, such as vomiting, nausea and diarrhea. Besides, all caregivers and mothers were informed that they can withdraw their children from the study at any time without giving reasons. 

The trial received ethical approval from the Faculty of Biological Science, the University of Dhaka, Bangladesh (Ref# 46 /Biol. Scs. /2017-2018), and the Griffith University Human Ethics Committee, Australia (Ref# 2017/467). The trial was registered with the International Standard Randomized Controlled Trial Register, number ISRCTN60058115.

### 2.2. Procedure

A site selection assessment was conducted in 6 sub-districts of the northern part of the country. The Belkuchi sub-district was selected because of the higher availability of eligible child–tube-well pairs (children of the stipulated age drinking from “high-iron” tube-wells). Screening was carried out in three unions (the lowest administrative division of the country, consisting of a cluster of villages—Belkuchi Pourashava, Bhangabari and Daulatpur—of the Belkuchi sub-district to identify the children (2–5 years old), who use the “high-iron” wells (≥2 mg/L) for drinking water. During the screening, 436 children from 8 villages were listed as potential participants. At the time of recruitment (roughly 2 months after the screening) for the study, 83 children were excluded as they did not meet the selection criteria. Besides, 10 subjects did not show up and another 16 subjects refused to take part in the study (Figure 1). Thus, the overall response rate was 92.6%. 

After obtaining either a signature or thumb impression on the written informed consent form of the parents/legal guardians for the participation of their wards in the trial, the mothers of all the enrolled children (*n* = 327) were interviewed using a structured questionnaire for baseline data collection. Blood samples were collected from sub-samples for assessing hemoglobin and the iron status parameters (Appendix A). The questionnaire consisted of several domains—socio-economics, child morbidities, as well as dietary and water intake assessments. Further, iron and arsenic levels of the drinking water were assessed. Socio-economic variables included household head’s occupation, mother’s education, spends on purchasing food, household food insecurity, ownership of assets, as well as types of household and toilet used. An asset index was constructed considering the socio-economic variables by using a principal component analysis [24,25]. Household food insecurity was assessed by calculating the HFIAS score based on three domains—anxiety of the impending insecurity, qualitative deprivation, and the quantitative deprivation of food intake [26]. We assessed dietary intake of the children using an interviewer-administered seven-day semi-quantitative food frequency questionnaire (SQFFQ), consisting of a list of commonly consumed local foods adapted from the national micronutrient survey 2011–2012 [20,27]. The SQFFQ was modified by adding some food items and validated against two 24 h recalls. The energy-adjusted correlation coefficient for iron was 0.60, *p* < 0.001, and the weighted kappa statistic was 0.30, falling within the acceptable range (unpublished). All the food items enlisted were assessed for the daily average intakes. The nutrient intakes were calculated using an updated Food Composition Table (FCT) on Bangladeshi foods [28]. For the foods which were missing in the FCT, the USDA database on the nutrient values was used [29]. Children’s body weight was measured using a bathroom scale (Tanita Inc., Japan) with a 100 g precision. The height was taken using a locally made wooden length board with a precision of 1 mm. The measurements were repeated, and the averages were considered. 

Venous blood samples (3.5 mL) were collected from the antecubital vein by a trained phlebotomist using a disposable syringe. An aliquot of the whole blood sample was taken in the EDTA tube for the measurement of hemoglobin and hemoglobinopathies. The remainder of the blood sample was dispensed in a centrifuge tube for collection of serum. The serum samples were transported to the laboratory in Dhaka city in an ice-gel cool box and stored at −70 °C until further analysis. Hemoglobin was measured by a Hemocue analyzer (Hemocue 301 Hemocue AB, Angleholm Sweden). Serum ferritin was measured using an electrochemiluminescence immunoassay (ECLIA) on an automated immunoassay analyzer (Cobas C311; Roche Diagnostics, Mannheim, Germany), using a commercial kit according to manufacturer’s instruction (Roche Diagnostics, GmbH, 68305 Mannheim, Germany). Serum TfR, serum CRP and AGP were determined by a particle-enhanced immunoturbidimetric assay on an automated, software-controlled clinical chemistry analyzer (Cobas c311, Roche Diagnostics GMBH, Mannheim 68305 Germany) using the commercial kits. The inter-assay coefficient of variations (CVs) for serum ferritin, sTfR, CRP and AGP were 0.32%–1.42%, 0.82%–1.14%, 3.6%–7.4% and 3.7%–6.5%, respectively. In the presence of inflammation or infection, serum ferritin concentration can be in the normal range or elevated despite deficient stores [30]. Thus, serum ferritin was adjusted for infection by using the raised values of CRP (>5 mg/L) and AGP (>1 g/L), by the correction factors calculated following Thurnham’s principle [31]. Congenital hemoglobinopathies, which is a potential confounder of hemoglobin status, were identified by capillary zone electrophoresis of Hb at pH 9.4 and a high voltage of 9600 V (Capillary 2 system; Sebia, Evry, France).

Iron concentration in the groundwater was measured using a Handheld Colorimeter (HI721 Checker^®^ HC (Hanna Inc. Woonsocket, RI, USA) with a range between 0.0 and 5.0 ppm, a resolution of 0.01 ppm, and an accuracy ±0.04 ppm ± 2% of the readings. Arsenic concentration in the water sample was assessed using an arsenic test kit device (Prerana Laboratories, India). The device was a test strip color-comparator instrument, using the principle of reduction of the inorganic arsenic compounds (As + 3 and As + 5) present in the groundwater sample [32]. 

During the two-month-long intervention, the field assistants visited each participant weekly to record compliance. On the first week, the field assistants provided 10 MNP sachets to the mothers of the participants to last until the next visit and explained to them how to consume the MNPs. Thereafter, in each weekly visit, the mothers were asked about the number of MNPs consumed by their children in the previous week. The actual consumption of MNPs was recorded after confirming by counting the returned empty and the intact sachets and replenished with MNPs to last another 10 days. Besides, the portion of the MNP-mixed food that was not consumed by the child was recorded. This information was considered while calculating the total iron consumption from MNP. The interviewers collected data on the episodes of various morbidities, such as diarrhea, loose stools, nausea, vomiting, fever, common cold and acute lower respiratory tract infection (ALRI) by asking mothers of the children every week. Loose stools implied the loose or watery consistency of stool. Diarrhea was defined as three or a higher number of loose, liquid or watery stools over 24 h, separated in time from an earlier or subsequent episode by at least 2 consecutive diarrhea-free days [33,34]. Fever was defined as an axillary temperature higher than 38.3 °C reported by the study worker who measured the temperature using a thermometer. The common cold was defined as cough, sneezing and fever, both without a rapid respiratory rate or chest in-drawing [35]. An acute lower respiratory infection was defined as cough or difficult-breathing, a rapid respiratory rate (>40 breaths per minute in children 12 months of age and older), and either a fever of >38.3 °C or chest retractions [35].

### 2.3. Statistical Analysis

#### 2.3.1. Non-Inferiority Margin and Sample Size

The treatment effects of a low-iron MNP were determined against the standard MNP using a non-inferior design. Typically, the non-inferiority margin is a fraction, a one-half or less of the historical effect size of standard treatment [36,37]. To determine the non-inferiority margin, we considered an earlier trial examining the efficacy of the standard MNP on hemoglobin status in rural anemic children of Bangladesh, which demonstrated an increase in hemoglobin level by 1.61 g/dL following an 8-week-long intervention [38]. Since all the children irrespective of anemia status were enrolled in the present study, we considered the non-inferiority margin of a modest 0.5 g/dL, which was roughly 30% of the effect size of the earlier study [38]. The non-inferiority of the low-iron MNP compared with the standard MNP for the effect on hemoglobin outcome was concluded if the lower bound of the one-sided 95% confidence interval for the treatment effect was higher than −0.5 g/dL.

We estimated the sample size for the hemoglobin outcome considering the non-inferiority margin. We assumed to establish with 95% confidence that the mean hemoglobin concentration in the low-iron MNP group would be no more than 0.5 g/dL lower than that in the standard MNP group. With the within-subject standard deviation of 1.1 g/dL [20], and using a one-tailed alpha of 5%, with 90% power, the required sample size per group was 83. Considering a 35% attrition due to follow up, 112 children were required in each group, and 224 in the two groups. The mean occurrence of diarrhea in an earlier Bangladeshi trial with the standard MNP was 1.17 cases per child over 3 months [39]. Since a low-dose of iron was used, we assumed that the low-iron MNP would result in a 30% lower magnitude of diarrhea than that reported elsewhere [39]. To detect a significant change in diarrheal incidence, with a 5% alpha and 80% power, as well as 35% attrition, the required sample size per group was 162; thus, a total of 324 samples was needed for two groups. 

#### 2.3.2. Statistical Analysis

All variables were checked by visual assessment of histograms and normality was tested by the Kolmogorov–Smirnov normality test. Model assumptions for the treatment effect of the low-iron MNP were tested for linearity by plotting residuals against the covariates. We examined the normality of the model by a kernel density plot of residuals, standardized normal probability (pnorm plots) and the quantiles of a variable against the quantiles of a normal distribution (qnorm plots). We plotted the residuals versus the fitted (predicted) values to assess the heteroskedasticity (Appendix A). The variance inflation factor was estimated to assess the multicollinearity of the model. The multivariable model seemed linear, had normally distributed residuals, homoskedasticity and showed no evidence of multicollinearity. 

Baseline characteristics of the households and study participants and the morbidity data of the children were presented as the mean ± SD for continuous variables and as percentages (*n* (%)) for categorical variables. Descriptive data were compared between the two MNP groups using the independent sample *t*-test for continuous variables and Chi-square test or the Fisher’s exact test with a two-sided significance level for categorical variables. Pearson’s correlation coefficients were estimated for determining the association between the total intake of iron and the body iron-reserve, after the logarithmic transformation of the relevant variables.

The treatment effects of the low-iron MNP on changes in the concentration of hemoglobin and the iron status markers against the standard MNP was compared by generalized linear modelling (GLM). The dependent variables were the changes (endpoint—baseline) in hemoglobin, ferritin and sTfR concentrations following the intervention; the treatment group was the independent variable. The covariates for adjustment were·(i) socio-economic variables (mother’s education, possession of cultivable lands, household food insecurity and spending on purchasing of food); and (ii)·the child’s characteristics (age, gender, thalassemia status, height-for-age *Z* score, baseline suffering of loose stools, baseline intakes of dietary and groundwater iron, iron intake from MNPs and the baseline values of the corresponding biochemical parameters). We employed the sandwich estimator of variance (i.e., robust standard error) to estimate unbiased standard errors for the effect estimates [40]. The treatment effects of the low-iron MNP against the standard MNP were reported as coefficients with robust standard errors with 95% confidence intervals. Intention-to-treat and per-protocol analyses were done to examine the treatment effects.

The Poisson regression model was used to compare the incidences of various morbidities between the two treatment groups and to report the comparative effect of the treatments as the incidence rate ratio. The dependent variables were the incidence of various morbidities (diarrhea, loose stool, nausea, vomiting, fever, common cold and ALRI) over the two-month-long intervention period. The treatment group was the independent variable, and the length of the exposure time, i.e., person-week for the morbidity conditions, was the exposure variable. We controlled for the socio-economic and child characteristics, which are prognostic to outcomes as covariates, as stated elsewhere. We also controlled for the mother’s hand-washing behavior and the duration that the child was breastfed, as the development of the immune system and breastfeeding are linked [41]. The incidence rate of the morbidities for the low-iron MNP relative to the standard MNP was considered significantly different when the incidence rate ratio (IRR) with 95% CI was estimated with a *p*-value < 0.05. Body iron-reserve was calculated after 2 months of intervention using Cook’s method [42] and compared between the treatment groups.

## 3. Results

During the screening in July–August 2018, 436 prospective children were listed, and 327 of them were enrolled for the study (Figure 1). During the intervention, 7 children refused in the standard MNP and 6 discontinued (4 refusals, 1 migration and 1 detected with the abdominal tumor) in the low-iron MNP group (Figure 1). The dropout rate was 4.26% and 3.68% in the groups, respectively. 

### 3.1. Baseline Household and Children Characteristics and Changes in Anemia, ID, Weight and Height of the Children Over the Intervention

Table 1 presents the household characteristics, e.g., socio-economics, food insecurity and iron concentration in the groundwater, which did not differ between the treatment groups. 

Mean iron concentrations ±SD of the groundwater were 8.22 ± 7.27 mg/L and 7.78± 7.51 mg/L in the standard MNP and the low-iron MNP groups, respectively. The groundwater in one tube-well contained an arsenic level of 10 ppm in the standard MNP group, while none of the tube-wells was detected with arsenic in the low-iron MNP group. On average, children were roughly 40 months old at baseline with no significant difference between the groups. The proportion of female was slightly higher in the low-iron MNP group, but the difference was not statistically significant (Table 1). 

The prevalence of anemia was 5.4% and 5.8% at baseline, and 1.0% and 2.5% after the intervention in the standard MNP and low-iron MNP groups respectively. The differences between the two groups were insignificant both at baseline and the end point (Table 2). Prevalence of hemoglobinopathies were 13% in each of the groups (Table 2).

The proportion of children with elevated AGP were 20.1% in the low-iron MNP group and 10.8% in the standard MNP group (*p* = 0.05). At baseline, iron deficiency (ID), based on infection-adjusted ferritin concentration, were 1.8% and 1.68%, respectively, and none of the children had an ID at end-point. Mean weight and height in the children increased significantly (*p* < 0.001) following the intervention in both the groups (Table 2).

### 3.2. Treatment Effect of the Low-Iron MNP on Hemoglobin and Iron Parameters

The GLM results showed that the low-iron MNP resulted in a 0.14 g/dL lower effect on the hemoglobin concentration compared with the standard MNP (β = −0.14, 95% CI: −0.30, 0.013; *p* = 0.07). The lower bound (−0.30 g/dL) of the 95% CI for the difference in the effect was higher than the priori non-inferior margin (−0.50 g/dL) (Table 3). 

There was no significant difference between the treatment effects of the groups in the infection-adjusted serum ferritin (β = 0.003, 95% CI: −6.31, 6.32; *p* = 0.99) and for serum transferrin receptor levels (β = −0.20, 95% CI: −0.44, 0.04; *p* = 0.09). 

### 3.3. Body-Iron Reserve, Daily Intake of Iron from All-Sources and Intake of Total Supplemental Iron from MNP Throughout the Intervention Period

After the two months of intervention, body iron reserve increased significantly in both the standard MNP (548.8 to 592.4 mg, *p* < 0.001) and in the low-iron MNP groups (569.8 to 614.5 mg, *p* < 0.001) (Table 4). 

Over the 2-month intervention period, the intakes of total supplemental iron were 633.6 ± 159.8 mg and 261.1 ± 55.1 mg in the standard and low-iron MNP groups, respectively (*p* < 0.001). The increase in the body-iron reserve from baseline to end-point were 7.94% and 7.84% in the standard MNP and in the low-iron MNP groups, respectively (*p* = 0.86; Table 4).

The daily total combined intake of iron from all sources (diet, groundwater and MNP) was higher in the standard MNP group than in the low-iron MNP group (18.25 mg vs. 12.37 mg; *p* < 0.001); the difference was largely attributed to the higher amount of iron from MNP (Appendix A). During the two months intervention period, after adjusting for actual intake of MNP-mixed food, the mean consumption of MNP sachets were 50.68 ±12.7 (standard MNP) vs. 52.22 ±11.0 (low-iron MNP), *p* > 0.05, which was 84.46% and 87.0% of the total allocated consumption, respectively (Appendix A).

### 3.4. Correlation between the Total Intake of Iron from All Sources and the Body-Iron Reserve

There was a moderate correlation between the total intake of iron from all sources and the body iron reserve in the low-iron MNP group (*r* = 0.28, *p* = 0.03); no such correlation was observed in the standard MNP group (Figure 2).

### 3.5. Morbidity Pattern by the Treatment Group and the Treatment Effect of the Low-Iron MNP on the Morbidities

During the 2-month intervention period, there were significantly fewer children in the low-iron MNP group that suffered from diarrhea than in the standard MNP group (14.8% vs. 23.1%; *p* = 0.05). The mean number of diarrhea (0.19 vs. 0.32; *p* = 0.05) and loose stool episodes (1.36 vs. 2.64; *p* = 0.008) were lower in the low-iron group (Table 5). No differences were observed between the groups in the occurrence and number of episodes of other morbidities and usage of medical treatment and consultations.

The trends of the weekly occurrences of loose stools remained higher in children receiving the standard MNP than in the children receiving the low-iron MNP (Figure 3). 

The results of the Poisson regression model indicated a significantly lower incidence rate of diarrhea in the low-iron MNP group compared with the standard MNP (IRR = 0.29, 95% CI: 0.11–0.77, *p* = 0.01). The incidence rate for loose stool was lower, approaching statistical significance in the low-iron MNP group (IRR = 0.46, 95%CI: 0.19–1.09, *p* = 0.08). We observed a significantly lower incidence of nausea (IRR = 0.24, 95% CI: 0.09–0.59, *p* = 0.002) and fever (IRR = 0.26, 95% CI: 0.15–0.43, *p* < 0.001) in the low-iron MNP group compared to the standard MNP group (Table 6). No differences in the incidence of other morbidities were observed between the groups.

## 4. Discussion

This randomized controlled trial examined the effect of a low-dose iron MNP against the standard MNP on hemoglobin and iron status in rural Bangladeshi children (2–5 years old), who drink from the “high-iron” groundwater. Using the intention-to-treat analysis, we observed, the lower bound of the 95% CI for the difference of the treatment effect of the low-iron MNP with the standard MNP was −0.3 g/dL. This was above the priori non-inferior margin of the acceptable difference of −0.5 g/dL, thus establishing the non-inferiority of the low-iron MNP against the standard treatment. The per-protocol analysis also yielded similar findings (Appendix A). The finding of the low baseline (5.4%–5.8%) and end-point (1%–2.5%) prevalence of anemia warrants discussion. Our study site resides in the areas with a very high concentration of iron in groundwater [17]. Of note, in the present study samples, the median value of iron concentration in groundwater was 4.54 mg/L (mean: ~8 mg/L), which was much higher than the cut-off for defining the “high” level of iron in groundwater [18]. There were hardly any children who were iron deficient (baseline < 2%, end-point 0%). Taking these into considerations, the low prevalence of anemia was not surprising. Further, we used a venous blood sample to measure hemoglobin concentration. Studies have shown that the capillary blood sampling, which is commonly employed for measuring hemoglobin concentration in surveys and studies, tend to overestimate anemia estimates [44,45,46]. The reasons for the difference between the methods are the measurement errors (mostly happens with capillary sampling) or the biological variability, which is difficult to minimize [44,46]. 

We observed the usage of low-iron MNP (5 mg iron) resulted in significantly fewer incidences of side-effects, such as diarrhea, nausea and fever, compared with the usage of the standard MNP (12.5 mg iron). The lower incidence of side-effects from a low-iron MNP is expected since these morbidities commonly occur with iron supplementation [5,6]. The findings of the low incidence of side-effects with the low-iron formulation are promising for the MNP programs in Bangladesh that suffers from suboptimum coverage, and side-effects were identified as an important underlying cause of the poor coverage [21].

Studies examining the efficacy and morbidities of the low-iron MNP are scarce. Samuel et al. [13], in Ethiopian children, have shown that a low-iron MNP containing 6 mg of iron in combination with an infant and young child feeding (IYCF) intervention effected in a marginal improvement of hemoglobin compared with the non-intervention group (no-iron), but caused a higher incidence of diarrhea [13]. This was relatively consistent with our finding, as we observed fewer incidences of diarrhea with the low-iron formulation compared with the standard MNP, which contain a higher amount of iron. Paganini et al. observed in young Kenyan infants that the MNP with 5 mg of highly bioavailable iron resulted in a 50% reduction in anemia over a 4-month intervention when compared with the control (no iron) [14]. These trials, e.g., Samuel et al. and Paganini et al., demonstrated the superior efficacy of the low-iron MNP (5–6 mg of iron) against the control (0.0 mg of iron) on hemoglobin concentration, while the present study showed a non-inferior efficacy of the low-iron MNP (5 mg of iron) against the standard MNP (12.5 mg of iron), which is a logical outcome. 

Paganini et al. employed a highly bioavailable iron in their low-iron formulation (containing 2.5 mg ferrous fumarate + 2.5 mg NaFeEDTA + 190 FTU phytase), and they observed an 18.8% absorption of iron [14]. In the present study, the low-iron MNP contained 5 mg of iron as ferrous fumarate. Tondeur et al. showed that ferrous fumarate in MNP, mixed in a cereal-based diet, had an absorption rate of 4.65% in the iron-replete children [47]. Using the ferrous fumarate and presumably with a much lower rate of absorption of iron than in Paganini et al.’s trial, the present study demonstrated the efficacy of low-iron MNP in preventing low hemoglobin levels, which could be explained by the consumption of iron from groundwater. Iron in groundwater remains mostly in a reducing and bioavailable (ferrous) state [15,48], and is reported to have a high absorption rate [49]. We considered the intake of iron from all sources—diet, groundwater and MNP—and calculated the amount of potentially bioavailable iron, considering the differential absorption potentials for different sources. Based on a study of the absorption of iron from iron-rich natural water [49], we assumed an estimated absorption potential for iron from groundwater. Accordingly, the estimated lowest amount of potentially bioavailable iron from all sources combined in children taking the low-iron MNP was 0.85 mg/day (Appendix A), which is sufficient to meet the daily requirement in this group of children [50].

The body iron reserve was sufficient, with >550 mg of baseline values in all groups. There was a similar magnitude of the increment of the iron reserve from baseline to end-point in both treatment groups, though the intake of supplemental iron in the standard MNP group was ~2.5 times (633.6 vs. 261.1 mg) higher than that in the low-iron MNP group. This suggests that, relative to the dose of iron, the amount of absorption of iron was smaller in the standard MNP group compared to its counterpart. This might have led to a higher amount of unabsorbed iron in the intestinal tract for the standard MNP group, which might have contributed to a significantly higher number of diarrheal and loose stool episodes observed in that group than in the low-iron MNP group. Further research is needed in this setting to examine the iron-induced adversities on the composition of gut microbiota, which is linked with iron supplementation and the occurrence of diarrhea and loose stool, to support the present findings. 

The combined intake of iron from all sources (diet + groundwater + MNP) were 18.25 and 12.37 mg in the standard MNP and the low-iron MNP groups, respectively (Appendix A). There was no group difference for intakes from dietary and groundwater sources; the difference was attributed to the intake of iron from the different MNPs. An intake of 18.25 mg iron from all sources did not show any association with the body-iron reserve in the standard MNP group (*r* = 0.02; *p* = 0.87). However, the intake of 12.37 mg iron in the low-iron MNP group showed a significant association (*r* = 0.28, *p* = 0.03). One possible explanation for the differential outcome between the groups is that the higher amount of iron in the standard MNP group might have initiated the stimulation of hepcidin at some point, through the iron-transferrin transportation complex [51]. This might have led to the subsequent inhibition of the absorption of further iron from the intestine [51], thus limiting the buildup of an iron reserve in the standard MNP group. This was reflected in the similarities of the levels of body-iron reserves between the groups at the end-point. However, for the low-iron MNP, a moderate degree of association indicates that the amount of iron (i.e., from low-iron MNP and other sources) present in the duodenum maintained a positive gradient of absorption of iron with minimal/no inhibition of absorption. This suggests that the dose of iron (5 mg) in the low-iron formulation was optimum in Bangladeshi children exposed to a high level of iron from groundwater. As the absorption was efficient, there might be less iron remaining unabsorbed, leading to lower incidences of side-effects (e.g., diarrhea, loose stool, nausea and fever) compared with the standard MNP group. This was further complemented by the findings of the mean number of loose stools by weeks during the 2-months intervention, which after initial occurrences in both the groups, declined and stabilized in the low-iron MNP group from the 4th week onwards. However, it continued to occur in higher numbers in the standard MNP group.

A baseline prevalence of ~5.5% anemia in a high iron groundwater area may question the relevance of the iron supplementation program for the prevention of anemia in children. However, the iron level in groundwater is considerably variable in the tube-wells [15,17]. In a predominantly high iron groundwater area, there are the wells that contain either no iron or a negligible level of iron (<0.3 mg/L, the WHO aesthetic limit) [52]. Hence, in the context of a less diversified traditional diet with suboptimum dietary iron [27], the absence of the supplementation program might be counterproductive to some children even in the high iron groundwater areas. In this setting, the low-iron MNP with a reduced risk of side-effects can be an optimum measure.

A limitation of the study was that one of the main investigators, who did the preliminary analyses, could not be blinded to the treatment group coding. This might have introduced some risk of bias. Unfortunately, this could not be avoided as the MNP preparations were imported from India and the customs clearance required the declaration of the composition of the different MNP preparations. However, all field personnel engaged in the distribution and recording of the compliance of MNP consumption and morbidity data, and parents of the children remained blind to the treatment group coding. Morbidity data were collected on the weekly recalls. The method, though widely practiced, is subject to recall bias. However, we provided extensive training to the monitoring staffs to collect data objectively. Among the strengths of the study, the uptake of the interventions was satisfactory (~86% MNPs were consumed) (Appendix A). Dropouts were fewer (<5% in the groups), which improved the precision of the findings. 

## 5. Conclusions

In conclusion, in Bangladeshi children, who are largely iron-replete from the source of drinking water, the low-iron MNP was efficacious in preventing low levels of hemoglobin compared with the standard MNP treatment. It resulted in a lower incidence of morbidities—diarrhea, nausea and fever—than the standard MNP. The low-iron MNP, being efficacious and safer, has a potential policy consideration for prevention of childhood anemia in Bangladesh, where groundwater iron level is predominantly high in many parts [17,20] and the coverage of the MNP program is suboptimum. The formulation can be evaluated for effectiveness and compliance in a program context operated in the high iron groundwater areas. Further research is needed to examine the efficacy and side-effects of the low-iron MNP in the predominantly low groundwater iron areas. Globally, in similar environmental settings, the findings may generate interests to assess the groundwater iron profile and exploring the optimum iron/MNP supplements for prevention of childhood anemia, as some 2 billion people, mostly in the low- and middle-income countries, rely on groundwater as potable supplies [53].

## Figures and Tables

**Figure 1 nutrients-11-02756-f001:**
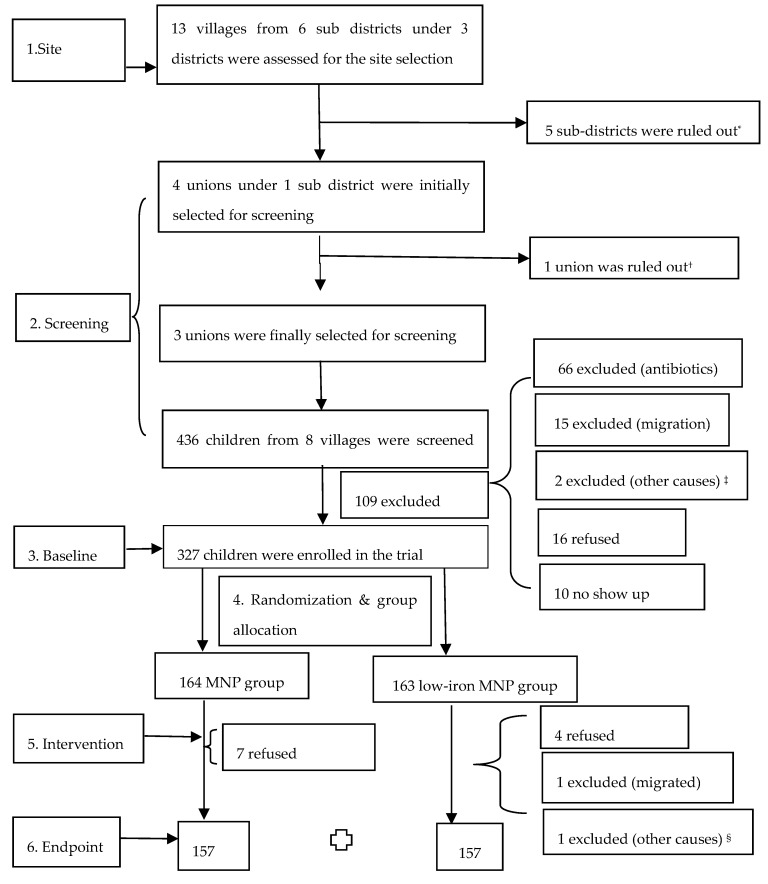
Selection process for the study participants. * Due to low availability of the child–well pair, and/or the logistical, geographical/natural calamity issues. ^†^ Due to an ongoing MNP program, which might have contaminated the study intervention. ^‡^ Not of the stipulated age (*n* = 1); a tumor in the abdomen (*n* = 1). ^§^ Diagnosed with a congenital neurological disease of the colon. As per the required sample size, roughly 70% of the enrolled children were randomly selected for assessment of the blood parameters.

**Figure 2 nutrients-11-02756-f002:**
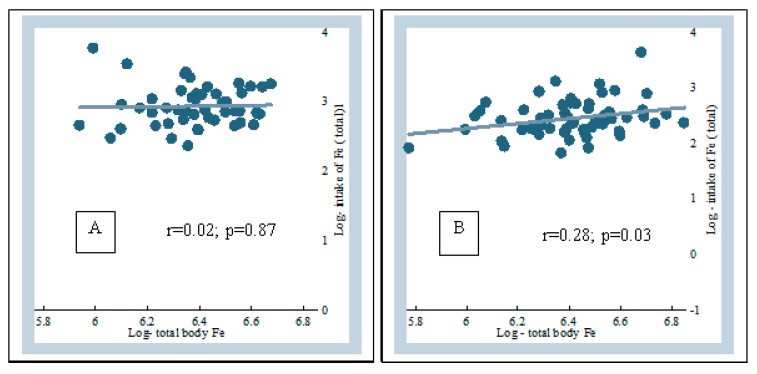
Correlation between the total intake *^,‡^ of iron and the body-iron-reserve ^†,‡^. * Total intake of iron was estimated by summing up the dietary iron, iron from MNP and the iron consumed from groundwater. ^†^ Total body iron was calculated using Cook’s method [42]. ^‡^ Total body iron and total intake of iron were log-transformed. (**A**) Standard MNP and (**B**) low-iron MNP.

**Figure 3 nutrients-11-02756-f003:**
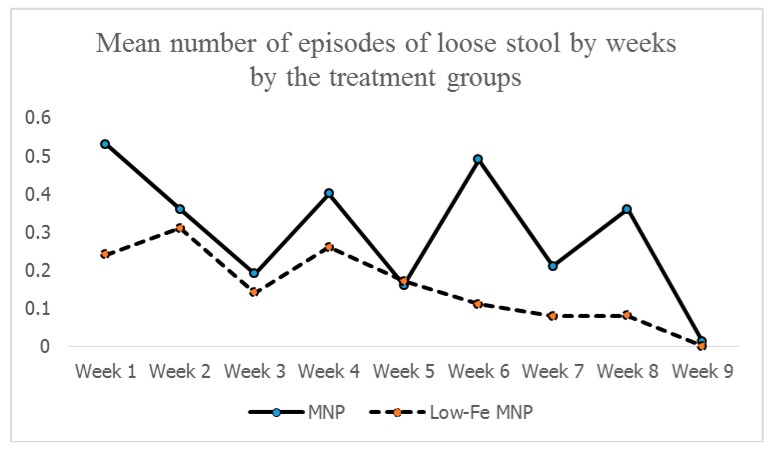
Mean number of episodes of loose stool by weeks * by the treatment group. * A total of 8 weeks and 4 days were required to complete the 60-day intervention, and thus it closes around the mid-point of Week 9.

**Table 1 nutrients-11-02756-t001:** Household and children characteristics at baseline by treatment group.

	Standard MNP		Low-Iron MNP		
Household Characteristics	Data Available	Data	Data Available	Data	*p*-Value
Occupation of household head *^¶^	164		163		
Business		32 (19.5)		28 (17.1)	0.41
Factory worker		46 (28.0)		56 (34.3)	
Unskilled laborer		23 (14.0)		25 (15.3)	
Farmer		21(12.8)		13 (8.0)	
Mother’s education (no of year) ^‡^	164	5.3 ± 3.3	163	5.3 ± 13.4	0.91
Possession of cultivable land *	164	53 (32.3)	163	51 (31.3)	0.84
Possession of improved housing *	164	43 (26.2)	163	40 (24.5)	0.08
Usage of unsanitary latrine *^,§^	164	32 (18.9)	163	27 (16.6)	0.26
Expenditure on food (BDT/week) ^‡^	164	1833.7 ± 881.4	163	1710.9 ± 739.9	0.17
Household food insecurity *^,‡‡^	164		163		
Food secure		77 (46.9)		84 (51.8)	0.45
Severe food insecure		17(10.4)		22 (13.6)	
Hand washing behavior of the mother	164		163		
Use soap before feeding child ^†^		28 (17.1)		34 (20.8)	0.71
Use soap after toilet ^†^		96 (58.5)		98 (60.5)	0.73
Iron concentration in groundwater (mg/L) ^‡^	164	8.2 ± 7.3	163	7.8 ± 7.5	0.59
Arsenic contamination of water (≥10 ppm)	140	1 (0.7)	140	0 (0.0)	n/a
Child characteristics					
Age (month) ^‡^	164	39.5 ± 9.1	163	40.2 ± 9.0	0.49
Gender female *	164	71 (43.3)	163	83 (50.9)	0.16
Breastfeeding	164		163		
Taken colostrum *		155 (95.1)		145 (90.1)	0.08
Exclusive breastfeeding *		7 (4.3)		11 (6.8)	0.31
Daily intake of ASF ^‡,^^║^ (gram raw-weight)	164	35.7 ± 29.3	163	37.3 ± 27.8	0.61

Data are reported as *n* (%) for the categorical variables, and as mean ± SD for the continuous variables. Group differences for the categorical variables were estimated by * Chi-square test or the ^†^ Fisher’s exact test as appropriate; group differences for the continuous variables were estimated by ^‡^ student’s *t*-tests. ^§^ Pit latrines without slab/open. ^║^ ASF includes small fish, large fish, eggs, chicken, beef, mutton and liver. ^¶^ The main occupations are presented. ^‡‡^ The severely food insecure and the food secure households are presented.

**Table 2 nutrients-11-02756-t002:** Differences in biochemical measures and anthropometry between the treatment groups at baseline and the end of the study period.

	Standard MNP		Low-Iron MNP		
Variable	Data Available	Data	Data Available	Data	*p*-Value
Anemia (hemoglobin < 11.0 g/dL)					
Baseline ^†^	111	6 (5.4)	120	7 (5.8)	1.0
End-point ^†^	103	1 (1.0) ^║^	116	3 (2.5) ^║^	0.62
Serum ferritin (ng/mL) ^§^					
Baseline ^‡^	111	67.0 ± 3.7	119	62.5 ± 2.6	0.73
End-point ^‡^	106	72.1 ± 3.2	115	69.7 ± 3.0	0.63
Iron deficiency (serum ferritin < 12.0 ng/mL)					
Baseline *	111	2 (1.8)	119	2 (1.7)	1.0
End-point	106	0 (0.0)	115	0 (0.0)	n/a
Serum TfR (µg/mL)					
Baseline ^‡^	47	3.99 ± 0.97	59	3.89 ± 1.0	0.59
End-point ^‡^	48	3.93 ± 1.02	58	3.64 ± 0.87	0.11
Iron deficiency (serum TfR > 8.3 µg/mL)					
Baseline	47	0 (0.0)	59	0 (0.0)	n/a
End-point	48	0 (0.0)	58	0 (0.0)	n/a
C-reactive protein (mg/L)					
Baseline ^‡^	111	1.7 ± 3.9	119	3.1 ± 9.1	0.13
End-point ^‡^	106	1.5 ± 3.5	115	1.9 ± 6.6	0.50
Alpha a1-acid glycoprotein (mg/dL)					
Baseline ^‡^	111	76.0 ± 29.5	119	75.8 ± 28.1	0.95
End-point^‡^	106	72.1 ± 25.8	115	74.1 ± 25.9	0.56
High C-reactive protein (CRP > 5 mg/L)					
Baseline *	111	8 (7.2)	119	16 (13.4)	0.12
End-point *	106	6 (5.6)	115	8(6.9)	0.69
High alpha a1-acid glycoprotein (AGP > 100 mg/dL)					
Baseline *	111	12 (10.8)	119	23 (20.1)	0.05
End-point *	106	14 (13.2)	115	15 (13.1)	0.97
Congenital hemoglobin disorders (any form)					
Present *	107	14 (13.1)	115	15 (13.0)	0.99
Helminth infestation					
Cyst of AL ^†^	51	1(1.96)	43	1 (2.32)	0.98
Cyst of Giardia ^†^	51	1 (1.96)	43	1 (2.32)	
Ova of AL *	51	7 (13.7)	43	6 (13.9)	0.97
Body weight (kg)					
Baseline ^‡^	164	12.38 ± 1.97	163	12.53 ± 2.0	0.49
End-point ^‡^	157	12.91 ± 2.05	157	12.9 ± 62.09	0.81
Height (cm)					
Baseline ^‡^	164	91.9 ± 6.91	163	92.42 ± 6.61	0.48
End-point ^‡^	157	93.29 ± 6.65	157	93.58 ± 6.76	0.70

Data are reported as *n* (%) for the categorical variables, and as mean ± SD for the continuous variables. Group differences for the categorical variables were estimated by * Chi-square test or the ^†^ Fisher’s two-sided exact test as appropriate; group differences for the continuous variables were estimated by the ^‡^ student’s *t*-tests. ^§^ Adjusted for high serum CRP and AGP according to Thurnham’s principle [31] ^║^
*p* > 0·05 between baseline and end-point. For reference body weight, 50^th^ percentile at the age of 24 and 59 months are 12.2 kg and 18.2 kg, respectively [43]. For the reference height, the 50^th^ percentile at the age of 24 and 59 months are 86.4 and 108.9 cm, respectively [43]. n/a: data not applicable.

**Table 3 nutrients-11-02756-t003:** Changes in hemoglobin and iron status markers, and comparative treatment effects of the low-iron Micronutrient Powder (MNP) vs. standard MNP.

Variable	Standard MNP		Low-Iron MNP		β (Robust SE)	95% CI	*p*-Value
	Mean	SE	Mean	SE			
Hemoglobin (g/dL)					Treatment effect *^,‡^ (Reference: standard MNP group)		
						*n* = 207	
Baseline	12.23 ^§^	0.07	12.37	0.07			
End line	12.46 ^║^	0.07	12.40	0.07			
Change^†^	0.23	0.07	0.032	0.06	−0.14 (0.08)	−0.30, 0.013	0.07
Serum ferritin (ng/mL)					Treatment effect *^,‡^ (Reference: standard MNP group)		
						*n* = 210	
Baseline	67.02 ^§^	3.70	62.48	2.64			
End line	72.15	3.22	69.68 ^║^	2.95			
Change ^†^	5.09	2.79	6.81	2.10	0.003 (3.22)	−6.31, 6.32	0.99
Serum TfR (µg/mL)					Treatment effect *^,‡^ (Reference: standard MNP group)		
						*n* = 104	
Baseline	3.99 ^§^	0.14	3.89	0.13			
End line	3.93	0.15	3.64 ^║^	0.11			
Change ^†^	−0.06	0.07	−0.20	0.09	−0.20 (0.12)	−0.44, 0.04	0.09

* Generalized linear model was used. ^†^ Changes in hemoglobin, ferritin and sTfR between end-point and baseline were the dependent variables; treatment group was the independent variable; the covariates for adjustment were: age, gender, thalassemia status, mother’s education; possession of cultivable lands; household food insecurity; spends on purchasing food; height-for-age *Z* score; baseline iron status markers depending on the type of the biomarkers analyzed; baseline morbidities, e.g., suffering from loose stools; baseline intake of dietary and groundwater iron; and the intake of iron from MNP. ^‡^ Intention-to-treat principle was applied. ^§^ The estimates were not statistically significantly different from the corresponding estimates of the other treatment group (*p* > 0.05). ^║^ The estimates were significantly different from the corresponding baseline estimates (*p* < 0.05). Unadjusted treatment effects; β: −0.19, 95% CI: −0.38, −0.01, *p* = 0.04 (hemoglobin); β: 1.71, 95% CI: −5.12, 8.55, *p* = 0.62 (serum ferritin); β: −0.13, 95% CI: −0.37, 0.10, *p* = 0.26 (sTfR).

**Table 4 nutrients-11-02756-t004:** Differences in the body-iron reserve and intake of iron from MNPs between the two treatment groups during the 2-month intervention.

	All		Standard MNP		Low-Iron MNP		
	Data Available	Data	Data Available	Data	Data Available	Data	*p*-Value
Body-iron reserve *							
Baseline (mg)	106	560.0 ± 117.4	47	548.8 ± 111.1	59	569.8 ±122.3	0.36
End-point(mg)	106	604.5 ± 113.9	48	592.4 ± 102.4	58	614.5 ± 122.6	0.32
Total iron intake from MNPs (mg) ^†^			164	633.6 ± 159.8	163	261.1 ± 55.1	<0.001
Increment ^‡^ of the body-iron reserve (%)	106	7.85	47	7.94	59	7.84	0.86

Data are reported as mean ± SD or % as appropriate. * Body iron reserve was estimated using Cook’s method [42]. ^†^ The intake of supplemental iron was calculated as the number of sachets (adjusted for actual intake of MNP-mixed food) consumed over two months (50.68 sachets: standard MNP; 52.21 sachets: low-iron MNP) multiplied by 12.5 (standard MNP) and 5.0 (low-iron MNP), respectively. ^‡^ The increment on the iron reserve (%) = (end line reserve—baseline reserve)/baseline reserve * 100.

**Table 5 nutrients-11-02756-t005:** Differences in common morbidities and medical treatment received during the 2-month intervention period between the children receiving the standard MNP and the low-iron MNP.

	Standard MNP		Low-Iron MNP		*p*-Value
Morbidities	Data Available	Data	Data Available	Data	
Suffered from loose stool *	160	48 (30.0)	162	35 (21.6)	0.08
No. of episodes ^†^	160	2.64 ± 5.17	162	1.36 ± 3.21	0.008
Suffered from diarrhea *	160	37 (23.1)	162	24 (14.8)	0.05
No. of episodes ^†^	160	0.32 ± 0.65	162	0.19 ± 0.50	0.05
Suffered from nausea *	160	35 (21.9)	162	32 (19.7)	0.63
No. of episodes ^†^	160	0.65 ± 1.53	162	0.51 ± 1.21	0.37
Suffered from vomiting *	160	51 (31.9)	162	55 (33.9)	0.69
No. of episodes ^†^	160	1.03 ± 2.0	162	0.87 ± 1.53	0.42
Suffered from fever *	160	103 (64.3)	162	99 (61.1)	0.54
No. of days ^†^	160	2.85 ± 3.23	162	2.97 ± 3.56	0.74
Suffered from common cold *	160	126 (78.7)	162	127 (78.3)	0.94
No. of days ^†^	160	7.75 ± 7.0	162	7.74 ± 7.12	0.99
Suffered from Acute Lower Respiratory Infection *	160	71 (44.4)	162	65 (40.1)	0.44
No. of days ^†^	160	2.02 ± 2.95	162	2.38 ± 4.02	0.35
Medical treatment					
Used Oral Rehydration Salt *	160	32 (20.0)	162	28 (17.2)	0.53
Used zinc *	160	8 (4.97)	162	7 (4.32)	0.78
Used antibiotics *	160	33 (20.6)		27 (16.6)	0.36
Consulted doctor *	160	94 (58.7)	162	91 (56.1)	0.64
No. of times consulted doctor ^†^	160	0.96 ± 1.07	162	0.99 ± 1.23	0.80
Needed referral to study physician *	160	55 (34.37)	162	52 (32.1)	0.66
No. of times needed referral to study physician ^†^	160	0.49 ± 0.77	162	0.41 ± 0.67	0.32
Needed hospital admission *	160	1 (0.63)	162	2 (1.2)	0.56

Data are reported as *n* (%) for the categorical variables, and as mean ± SD for the continuous variables. Group differences were estimated by * Chi-square test for the categorical variables, and by ^†^ student’s *t*-tests for the continuous variables. An episode of diarrhea was defined as three or more loose/watery stools over 24 h. To define another episode, at least a 48 h symptom-free interval was considered [33,34]. A fever was defined as an axillary temperature higher than 38.3 °C [35], measured by the field worker using a thermometer. A common cold was defined as cough, sneezing and fever (implying pharyngitis or rhinitis), without a rapid respiratory rate or the chest in-drawing [35]. Acute lower respiratory infection was defined as cough or difficulty breathing, a rapid respiratory rate (>40 breaths per minute in children 12 months of age and older) and either a fever of >38.3 °C or chest retractions [35].

**Table 6 nutrients-11-02756-t006:** Poisson regression modeling *^,†,‡^ to estimate the incidence rate ratio (IRR) of various morbidities in children for the usage of the low-iron MNP to the standard MNP over the intervention period.

Morbidities (Ref: Standard MNP)	IRR	Robust SE	95% CI	*p*-Value
Diarrhea	0.29	0.14	0.11–0.77	0.01
Loose stool	0.46	0.20	0.19–1.09	0.08
Nausea	0.24	0.11	0.09–0.59	0.002
Vomiting	0.63	0.32	0.23–1.71	0.36
Fever	0.26	0.06	0.15–0.43	<0.001
Common cold	0.77	0.28	0.37–1.61	0.49
ALRI	0.64	0.30	0.25–1.62	0.35

* Poisson regression was used to estimate the IRR of the morbidities in children receiving the low-iron-MNP and the standard MNP, considering the first occurrence of the event and the total exposure time of the condition (person-week). ^†^ Intention-to-treat analysis was applied. ^‡^ Adjusted for the baseline covariates (household and child characteristics). Unadjusted treatment effects IRR: 0.61, 95% CI: 0.36, 1.03, *p* = 0.06 (diarrhea); IRR: 0.70, 95% CI: 0.45, 1.08, *p* = 0.11 (loose stool); IRR: 0.86, 95% CI: 0.55, 1.35, *p* = 0.53 (nausea); IRR: 1.10, 95% CI: 0.75, 1.61, *p* = 0.61 (vomiting); IRR: 0.91, 95% CI: 0.75, 1.10, *p* = 0.33 (fever); IRR: 1.05, 95% CI: 0.81, 1.36, *p* = 0.69 (common cold); IRR: 0.91, 95% CI: 0.65, 1.27, *p* = 0.60 (ALRI).

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
