# Peer review of "Effect of Micronutrient Powder (MNP) with a Low-Dose of Iron on Hemoglobin and Iron Biomarkers, and Its Effect on Morbidities in Rural Bangladeshi Children Drinking Groundwater with a High-Level of Iron: A Randomized Controlled Trial"

_nutrients, 2019, doi:10.3390/nu11112756_

Round 1

Reviewer 1 Report

The main research findings of this paper will be important for applied nutrition of children in developing countries. The RCT seem to have been well designed and fully conducted, the data carefully analyzed, and the manuscript is well written.

Specific comments
1. Lines 100-102 How did you get consent? In writing? How did you provide information about side effect?
Please mention how did you get informed consent in more detail.
2. Figure 3. There is no episode of loose stool during week 9 in both treatment group.
This RCT includes two month intervention. Did week 9 mean post-intervention period?
Please clarify about this.
3. Table 5. It is difficult to read because the morbidities items and numbers are not aligned.
Please modify the Table.

Author Response

Reviewer#1

Comments and Suggestions for Authors

The main research findings of this paper will be important for applied nutrition of children in developing countries. The RCT seem to have been well designed and fully conducted, the data carefully analyzed, and the manuscript is well written.

Specific comments

Lines 100-102 How did you get consent? In writing? How did you provide information about side effect? Please mention how did you get informed consent in more detail.

Response: Thanks, we have now clarified (please see line 100-103 and line – 119).

Figure 3. There is no episode of loose stool during week 9 in both treatment group.

This RCT includes two-month intervention. Did week 9 mean post-intervention period?

Please clarify about this.

Response: Week 9 did not refer to a post-intervention period. The intervention was designed to feed the children 60 sachets of MNP over two months (1 sachet/day). To feed 60 sachets, 8 weeks and 4 days were required; hence the 60-day intervention essentially wrapped up in the 9th week. A footnote is now added in figure 3 to clarify the issue (line 387-388).

Table 5. It is difficult to read because the morbidities items and numbers are not aligned.

Please modify the Table.

Response: Thanks, Table 5 has now been formatted as suggested.

Reviewer 2 Report

This is a careful investigation with one important limitation: the results are only valid for regions with a high iron concentration in ground water. This is well recognised by the authors as mentioned in the conclusion. Th Hb-values used for definition of anaemia for the age-group studied should be mentioned in the abstract and introduction. Reference 8 seems not to be the correct one for the statement in lines 54-56. It is strange that no placebo arm was considered (line 88-89) for ethical reasons. The duration of the study was rather short, and by giving a placebo dose the children would not have suffered in any way. Differences between groups were so small that they had no clinical consequences. Line 122: consists > consisted. Line 129: two times 26. The list of abbreviations is not complete. In Table 2 normal range should be included for given values.

Author Response

Reviewer#2

Comments and Suggestions for Authors

This is a careful investigation with one important limitation: the results are only valid for regions with a high iron concentration in ground water. This is well recognised by the authors as mentioned in the conclusion.

Response: Thanks.

Th Hb-values used for definition of anaemia for the age-group studied should be mentioned in the abstract and introduction.

Response: Thanks, now added. Please see lines 29-30 (Abstract) and lines 41-42 (Introduction).

Reference 8 seems not to be the correct one for the statement in lines 54-56.

Response: Thanks. Yes, we agree with the reviewer. We have now rewritten the sentence to make a clear statement about reference 8 (Please see lines 51-52).

It is strange that no placebo arm was considered (line 88-89) for ethical reasons. The duration of the study was rather short, and by giving a placebo dose the children would not have suffered in any way.

Response: Thanks for the comment. Please note that the standard MNP formulation has been recommended by the Bangladesh Government for the prevention and control of anemia in children and accordingly, there is a significant presence of distributions of MNP by national NGOs. Thus, we did not consider a placebo arm due to ethical concern. This statement has now been added to the text (Please see line 85-88). However, we agree with the reviewer that by adding a placebo arm for such a short time may not have any significant adverse impact on the children.

Differences between groups were so small that they had no clinical consequences.

Response: Thanks. This depends on how we look at these results. First, we did not expect much changes in the hemoglobin/iron status but was hoping to reduce side-effects, especially episodes of diarrhea and loose stool. The latter findings may have important programmatic implication (i.e. increasing compliance and thus, better program impact).  

Line 122: consists > consisted.

Response: Thanks, now corrected (see line 123)

Line 129: two times 26.

Response: Thanks, now corrected (see line 130)

The list of abbreviations is not complete.

Response: Thanks, we have now added more abbreviations to complete the list (see page 16: list of abbreviations).

In Table 2 normal range should be included for given values.

Response: Thanks. Please note that there is a lack of the credible reference values for hemoglobin for the children, and among the available references, the ranges are highly variable. The WHO did not provide a range of normal hemoglobin levels, but the cut-off of defining anemia. Also, we did not find reference values for serum ferritin and sTfR in this age group (24-59 months). Hence, providing a normal range in this particular age group for the above parameters is not possible. However, we have provided the 50th percentile for body weight and height of the children aged 24-59 months as a standard for comparison (WHO 2006) as a footnote under the Table 2. (please see lines 293-296).

Reviewer 3 Report

The manuscript is a well written report of the results from a randomized controlled trial focused on evaluating an alternative for iron supplementation in specific areas that could potentially reduce side effects and consequently potentially improved uptake of iron supplementation. Considering the high global burden of anemia and its consequences this report could provide valuable information for the public health nutrition arena. 

Below I include some suggestions and/or comments for the authors that could be useful to improve the quality and readability of the manuscript.

Consider rewording the title, either by changing the word “Efficacy” for "Effect", or indicate what the efficacy of supplements refer to (e.g. to reduce anemia? to increase Hb and iron biomarkers?) The superscripts associating authors with affiliations need to be fixed Line 117: based on my calculations the response rate was 92.6% Line 123: based on the way is currently written it would seem that the iron/arsenic level in water was asked to parents Line 415: The design of the trials mentioned are not at all comparable with the one evaluated in this study. Lines 472-473: Was this investigator involved in preliminary data analysis? If so, then yes, that may have increased the risk of potential bias during analysis. Otherwise, the risk of bias was low. Table 2: is a sub-title row missing for Ferritin (ng/ml)? as done for CRP and AGP Table 3: a) please the sample size for each group and outcome reported, b) since this was a RCT, why did the authors decided not to present the unadjusted results? Per CONSORT guidelines for RCTs (Moher et al 2010), adjusted analyses are considered ancillary while the main results are those from unadjusted analyses (this comment applies also to Table 6), c) with regard to the SEs for Hb, consistency in number of decimals Table 5: Would it make sense to move the # of loose stools from Table 4 to this table? Appendix A: a) should the reader assumed that all biomarker sub-samples were selected randomly? b) what was the rationale for selecting a much smaller sample for sTfR?

Author Response

Reviewer#3

Comments and Suggestions for Authors

The manuscript is a well written report of the results from a randomized controlled trial focused on evaluating an alternative for iron supplementation in specific areas that could potentially reduce side effects and consequently potentially improved uptake of iron supplementation. Considering the high global burden of anemia and its consequences this report could provide valuable information for the public health nutrition arena. Below I include some suggestions and/or comments for the authors that could be useful to improve the quality and readability of the manuscript.

Consider rewording the title, either by changing the word “Efficacy” for "Effect", or indicate what the efficacy of supplements refer to (e.g. to reduce anemia? to increase Hb and iron biomarkers?)

Responses: Thanks, we now changed the word efficacy for effect – in the title (line 2), also in the abstract (line 22) and introduction (line 70) to be consistent.

The superscripts associating authors with affiliations need to be fixed Line 117:

Responses: The superscripts for the author’s affiliation have now been fixed (line 9-17).

based on my calculations the response rate was 92.6%

Responses: Thanks for identifying the mistake- now corrected (line 118)

Line 123: based on the way is currently written it would seem that the iron/arsenic level in water was asked to parents

Response: Thanks, we agree with the reviewer. Thus, made necessary adjustment (line 124-125)

Line 415: The design of the trials mentioned are not at all comparable with the one evaluated in this study.

Response: Thanks, we agree; design of the present trial (non-inferior) which is different from other relevant trials discussed (superiority trials). This has already been mentioned in the text (lines 450-453).

Lines 472-473: Was this investigator involved in preliminary data analysis? If so, then yes, that may have increased the risk of potential bias during analysis. Otherwise, the risk of bias was low.

Response: Thanks, we have now cleared this limitation more clearly. (please see lines 511-517).

Table 2: is a sub-title row missing for Ferritin (ng/ml)? as done for CRP and AGP

Response: Thanks, a subtitle row is added for ferritin (ng/ml)(See table 2)

Table 3: a) please the sample size for each group and outcome reported,

Response: Sample number is provided by groups about the outcome reported (See table 3).

b) since this was a RCT, why did the authors decided not to present the unadjusted results? Per CONSORT guidelines for RCTs (Moher et al 2010), adjusted analyses are considered ancillary while the main results are those from unadjusted analyses (this comment applies also to Table 6),

Response: We agree with the reviewer’s comment about the unadjusted analysis. However, to avoid the tables, look complicated, the findings of unadjusted analyses have now been added as additional footnotes to Table 3 and 6. (please see lines- 319-321 and lines 411-415)

Further, the treatment effects adjusted for the prognostic covariates are featured in tables as the adjustment improved the models and the precision of the estimates.

c) with regard to the SEs for Hb, consistency in number of decimals

Response: Thanks, the decimal point is now standardized to two spaces.

Table 5: Would it make sense to move the # of loose stools from Table 4 to this table?

Response: Thanks for identifying this. We now removed the # of loose stools from Table 4. This information is already presented in Table 5, as one of the key morbidities.

Appendix A: a) should the reader assumed that all biomarker sub-samples were selected randomly? b) what was the rationale for selecting a much smaller sample for sTfR?

Response: Yes, the biomarkers were sub-sampled randomly out of the enrolled children as per the required sample size. Regarding the sTfR, it was not the principal iron biomarker considered for assessment. The reason for considering a much smaller sample for sTfR was the constraint of funding.
